# Use of an inertial sensor and a force platform to assess static balance in participants affected by multibacillary leprosy

**Aymee Lobato Brito[1,2], Amelia Pasqual Marques[3], Yuzo Igarashi[1,4], Luana Karine Resende Oliveira[1,5], Luciane Lobato Sobral[6], Marília Brasil Xavier[4], Givago Silva Souza[2,4], Bianca Callegari** [1,4] *

**1** Laboratory of Human Motricity Studies, Federal University of Para, Belém, Pará, Brasil, **2** Biological Sciences Institute, Federal University of Para, Belém, Pará, Brazil, **3** Department of Physiotherapy, Speech Therapy and Occupational Therapy, Faculty of Medicine, University of São Paulo, Cipotânea Street, University City, São Paulo, Brazil, **4** Tropical Medicine Center, Federal University of Para, Belém, Pará, Brazil, **5** Association of Social Pioneers, SMHS Neighborhood, Brasília, Distrito Federal, Brazil, **6** Center for Biological and Health Sciences, State University of Pará, Belém, Pará, Brazil

* callegaribi@uol.com.br

**Data Availability Statement:** All data are in the manuscript and/or supporting information files.

**Funding:** 3. This work was funded by Universidade Federal do Pará in PAPQ program. GSS is CNPq

## Abstract

### Introduction

Leprosy is a chronic, slowly developing infectious disease that affects the peripheral nerves, specifically Schwann cells. Individuals with the multibacillary type exhibit a propensity for developing chronic pain and a decrease in sensitivity in the plantar region, which directly interferes with balance maintenance. The evaluation of static balance in this population is made through the measurement of the center of pressure (COP) oscillations. Therefore, there is a need to investigate the association between postural control and COP oscillations using a force platform and finding accelerations of the center of mass (COM) from inertial sensors for reliable and portable balance assessment in leprosy patients.

### Objective

To validate the application of inertial sensors for patients with leprosy by establishing a correlation with the outcomes obtained from a force platform.

### Methods

This is an observational study with a case-control design, in which 30 participants with leprosy and 30 healthy participants were recruited to evaluate static balance using an inertial sensor and a force platform. Participants underwent balance assessment under two conditions (Eyes Open: OE and Eyes Closed: CE), and data from the platform and sensor were processed using Matlab computational routines. The data were quantified using four parameters: Total Displacement (TD), Area, Antero-Posterior Displacement (APdisp), and Medio-Lateral Displacement (MLdisp).

productivity fellow (#309936/2022-5). BC is post-doc CNPQ fellow (#102167/2022-2). The funders had no role in study design, data collection and analysis, decision to publish, or preparation of the manuscript.

**Competing interests:** The authors have declared that no competing interests exist.

## Results

The evaluated parameters showed significantly different values between the groups, where the Leprosy group exhibited significantly higher values compared to the control group, both in the OE and CE conditions for all four parameters. The sensor corroborated the differences demonstrated by the platform and followed the same trend for medio-lateral displacements and accelerations. It can be observed that the evaluated parameters exhibited a varied correlation ranging from moderate to large between the platform and the sensor. Among the four variables, MLdisp had the lowest correlation.

## Discussion

The results partially confirmed the first hypothesis of concurrent validation, showing a moderate to large correlation between the force platform and the inertial sensor. The second hypothesis of clinical validation was also partially confirmed, as not all group differences observed in the COP measurements from the force platform were reflected in the COM measurements from the inertial sensor. Specifically, the force platform indicated greater oscillations in participants with multibacillary leprosy compared to controls, a finding statistically confirmed by the sensor for all measures except $ML_{disp}$.

## Conclusion

This research confirmed the concurrent validity of the inertial sensor with the force platform and its clinical validation, demonstrating that this instrument can be applied in clinical settings due to its low cost and ease of use. The findings may contribute to public health by identifying postural control tools for patients with multibacillary leprosy.

### Author summary

Our study aimed to validate the application of inertial sensors for patients with leprosy by establishing a correlation with the outcomes obtained from a force platform. Due to the disease's impact on sensory perception and chronic pain in the plantar region, maintaining balance becomes challenging for affected individuals. Using a force platform and an inertial sensor, we evaluated balance in 30 leprosy patients and 30 healthy participants under Eyes Open and Eyes Closed conditions. Results revealed significantly higher values in balance parameters for the leprosy group compared to controls, indicating greater instability. The inertial sensor confirmed these differences, particularly in medio-lateral displacements and accelerations. While correlation between the platform and sensor varied, our study validated the sensor's concurrent and clinical efficacy. This research underscores the potential of inertial sensors as reliable, cost-effective tools for assessing balance in clinical settings, especially for patients with multibacillary leprosy. Our findings contribute to enhancing public health efforts by offering practical solutions for monitoring and managing balance deficits in this neglected disease population.

## 1. Introduction

Leprosy is a neglected disease and it affects 200 000 new cases every year and it occurs in more than 120 countries around the world [1] caused by the bacterium Mycobacterium leprae, is a chronic infection characterized by a slow progression. The disease has the capability to impact both the skin and nerves, posing a potential threat to the overall health of affected individuals. In instances of severe cases, leprosy can result in significant consequences, such as loss of skin sensitivity, deformities and physical disabilities [1–3].

The disease has variability of clinical manifestation related to bacillary load presented by people infected by the *M. leprae*. The form of the disease with patients with low bacillary load is referred as paucibacillary, while that with patients with high bacillary loads is referred as multibacillary [2]. Multibacillary forms of the disease trend to elicit more skin lesions and marked peripheral nerve damages [3].

Individuals with multibacillary leprosy exhibit a higher propensity for developing chronic pain due to their involvement of a greater number of peripheral nerves [4,5], as well as a decrease in sensitivity in the plantar region, which directly interferes with increased plantar pressure and balance maintenance [6,7]. It was previously demonstrated that weight-bearing in these patients primarily occurs in the forefoot region compared to those with preserved local sensitivity, indicating that the loss of protective sensitivity contributes to higher plantar pressures and increased COP oscillations [8]. Alamino et al. [9] observed greater oscillations and velocity of COP in individuals with leprosy and Cordeiro et al. [10] observed that leprosy group showed greatest variation regarding the average values for the projection of the COP.

In addition to the established link between sensory loss and impaired balance control, the literature also describes the concept of "silent neuropathy" or "quiet nerve paralysis." This condition is characterized by the impairment of sensory and motor functions in the absence of the typical clinical signs of leprosy, such as skin lesions, nerve tenderness, pain, or paraesthesia. This suggests that significant nerve damage and dysfunction can occur even when clinical symptoms are not overtly present. Therefore, the early identification of neurological alterations in patients with leprosy is crucial for monitoring sensory and motor deficits, thereby preventing the progression to more severe disabilities associated with the disease [11].

Despite these findings, the evaluation of static balance in this population, through the measurement of center of pressure COP oscillations, is still scarce. In fact, balance assessment is not included in the standardized protocols established by the World Health Organization (WHO) (Brandsma & Van Brakel; 2004). The force platform is considered the gold standard for measuring balance and demonstrates good to excellent reliability in registering postural oscillation [12–14]. However, this equipment is costly and difficult to transport, which makes it impractical for use in field settings, especially in the primary attention [15]. These limitations could be reduced with the use of new portable digital technologies to objectively measure different aspects of balance.

The inertial sensors are innovative technological tools for capturing and analyzing motion [16], which shows excellent potential for assessing balance in remote rehabilitation environments [17,18]. It is a low-cost, portable, non-invasive instrument with high precision that can be attached to subjects or equipment to measure postural control [12,19,20]. As far as we know, this is the first study that applied this type of sensor in the assessment of the static balance of patients with leprosy. We hypothesized that COP and COM acceleration and displacement would be higher in patients with leprosy disease, and the sensor and platform would indicate group differences in a correlated manner. Thus, in this study our objective is aimed to validate the application of inertial sensors for patients with leprosy by establishing a correlation with the outcomes obtained from a force platform.

## 2. Methods

### 2.1. Ethics statement

The study was approved by the Research Ethics Committee of the Tropical Medicine Center of the Federal University of Pará—UFPA (report #5.468.074). Written informed consent was obtained from all participants prior to the start of the study.

### 2.2. Design and sample

An observational study with a case-control design was conducted to evaluate the static balance of healthy adult participants and those affected by multibacillary leprosy.

### 2.3. Population characteristics

A total of 30 participants with multibacillary leprosy (n = 21 males, 9 females, mean age 41.5 ± 14.7 years) and 30 healthy participants (n = 8 males, 22 females, mean age 42.5 ± 10.8 years) were selected. The participants from both groups reported not engaging in any form of physical activity during the assessment.

Healthy participants had no clinical history of leprosy and any other disease that could impair the balance control or even complains about balance impairment. Individuals with multibacillary virchowian leprosy were recruited from the outpatient clinic of the Tropical Medicine Center at the Federal University of Pará, where they were diagnosed and undergoing treatment. The patients had grade 0 disability and without deformities, as verified in the evaluation records and confirmed by the researchers in a subsequent assessment,' adults over 18 years of age, both male and female, and with sufficient understanding to perform the research procedures were included in the study. Individuals with other neurological disorders, presence of contagious dermatological lesions, lower limb amputations, and those unable to stand without the aid of assistive devices were excluded from the study.

### 2.4. Protocol

For the assessment of static balance, a force platform BIOMEC400 model (EMG System do Brasil, Ltda., SP, Brazil) and an inertial sensor (MetamotionC model, MbientLab, San Francisco, USA) were used. The lighting and sound conditions at the testing site were kept constant, and the researcher remained close to the participants throughout the test to prevent falls.

To begin the test, a sensor was attached to the participant's lower back using a tape, specifically at the 5th lumbar vertebra (L5), as it is close to the body's center of mass [21]. The sensor is a wearable commercial device, containing an embedded triaxial (X, Y and Z axes) accelerometer to measure and record the center of mass (COM) accelerations. This is a 25 mm in diameter x 4 mm ultralight device weighing only 5.6 g, with a replaceable 200 mAH battery with data transfer via a Low-Energy Smart Bluetooth and requires an Android or iOS application (MetaBase, MbientLab, San Francisco, CA, USA) to store the collected data, with a sampling rate set at 100Hz. The sensor used had been validated in a study assessing static balance in healthy participants [22]

The force platform used for participant positioning during the balance test was connected to the notebook with the Biomec software (EMG System do Brasil, Ltda., SP, Brazil) which was used to receive the platform's readings, with a sampling rate of 100 Hz.

Participants were positioned on the platform with bare feet, parallel stance, and hands by their sides. Two stages were performed to assess static balance, each lasting 60 seconds: In the first stage, the participant maintained a bipodal stance on the platform with open eyes (OE), fixed on a point one meter away. In the second stage, they closed their eyes (CE), maintained a

bipodal stance. They were instructed to jump vertically on the platform, previously to these two stages to synchronize the signals of both assessment instruments.

## 2.5. Data analysis

The data was exported as text files and subsequently processed using custom computational routines developed in Matlab (version R2020a, Mathworks, USA).

The time series were quantified using four selected parameters calculated using Matlab commands to analyze postural oscillations based on the signals from the equipments: Antero-posterior displacement/ acceleration ($AP_{disp}$), medio-lateral displacement/ acceleration ($ML_{disp}$), total displacement/ acceleration (TD), and area (AR).

i. Antero-posterior and medio-lateral displacements ($AP_{disp}$ and $ML_{disp}$) were calculated from the root mean square (RMS) amplitude of the stabilograms in the medio-lateral and antero-posterior axes, as described in Eq 1. For the force platform, this parameter was quantified in centimeters and for the sensor this parameter was quantified in meters per second square.

$$\text{Amplitude RMS} = \sqrt[2]{\frac{\sum_{i=1}^{N}(x_i)^2}{N}} \tag{1}$$

Where X is the value of the device reading at recording time $i$, and n is the total number of readings in the anteroposterior and medio-lateral axes.

ii. Total displacement (TD) of the stabilogram represents the length of the center of pressure/ center of mass trajectory on the base of support and is represented by Eq 2. For the force platform, this parameter was quantified in centimeters and for the sensor this parameter was quantified in meters per second square.

$$TD = \sum_{i=1}^{N} \sqrt[2]{AP_i^2 + ML_i^2} \tag{2}$$

Where AP and ML are the values of the readings at temporal point i in the antero-posterior and medio-lateral displacements/acceleration, respectively; μ is the mean of the readings within the range of interest. N is the total number of temporal points in the measurement.

iii. The area (AR) of the stabilogram displacement is represented by Eqs 3 and 4. For the force platform, this parameter was quantified in centimeters and for the sensor this parameter was quantified in meters per second square.

$$[vec, val] = eig(cov(AP, ML)) \tag{3}$$

$$\acute{A}rea = pi*prod(2.4478 \times \sqrt{svd(val)}) \tag{4}$$

Where vec and val are the eigenvectors and eigenvalues of the covariance matrix of the antero-posterior and medio-lateral oscillations/ acceleration, respectively.

## 2.6. Statistical analysis

The sample size calculation was performed using G*Power software, version 3.1.9.4, based on the independent two-group t-test, considering an effect size of 0.97, a power of 95%, and a significance level of 5%. The calculated sample size (n) was determined to be 29 for each group, resulting in a total of 58 participants.

Statistical analysis was performed using the 'IBM SPSS Statistics for Windows, version 25 (IBM Corp., Armonk, N.Y., USA), and the data were tested for normality using the Shapiro-Wilk test, which indicated a non-normal distribution. Mann Whitney was used to compare the variables between the groups and to assess the correlation between the instruments the Spearman correlation coefficient was employed, which was interpreted with a magnitude as follows: 0–0.1: trivial; 0.1–0.3: small; 0.3–0.5: moderate; 0.5–0.7: large; 0.7–0.9: very large; 0.9–1.0: almost perfect [23]. A confidence level of 5% was considered for all statistical procedures.

## 3. Results

### 3.1. Sample characteristics

Table 1 presents data concerning the participants' characteristics, illustrating the lack of significant differences in height, age and weight between the groups. Furthermore, it furnishes details regarding the clinical assessment of leprosy patients.

### 3.2. Balance evaluation

Fig 1 depicts the stabilograms derived from recorded data acquired from a representative subject of both the Leprosy and control groups via the sensor (A) and force platform (B) under conditions of open eyes (OE) and closed eyes (CE). It is notable that the $AP_{disp}$ parameter demonstrates larger amplitudes, in comparison to $ML_{disp}$, under both OE and CE conditions for both groups and instruments. Additionally, the Leprosy group displays greater amplitudes across all parameters when contrasted with the control group.

Fig 2 displays the inertial sensor statokinesiogram (A) and force platform statokinesiogram (B) of a representative subject from each group during the examination conducted under conditions: Open eyes (OE) and closed eyes (CE). $AP_{disp}$ and $ML_{disp}$ displacements are represented. The control group is denoted as A, while the Leprosy group is denoted as B. ACC refers to the accelerometer.

The medians and quartiles are presented in Table 2. Significant differences were observed in all four parameters measured by the force platform between the groups, with the Leprosy group exhibiting significantly higher values compared to the control group, in both the OE and CE conditions. The sensor corroborated the differences demonstrated by the platform, except for mediolateral analysis.

### 3.3. Sensor validity

Fig 3 illustrates the correlation between measurements obtained by the force platform (x-axis) and the inertial sensor (y-axis). It is evident that the assessed parameters displayed diverse correlations, ranging from moderate to large, between the two instruments.

**Table 1. Demographical features of the sample.**

|  | Leprosy | Control | p-value |
|---|---|---|---|
| **Number of participants** | 30 | 30 |  |
| **Sex** (Male/female) | 21/9 | 8/22 | 0.87 |
| **Age (yrs)** | 41.5 ± 14.7 | 42.5 ± 10.8 | 0.73 |
| **Weight (kg)** | 70.8 ± 13.8 | 67.5 ± 11.1 | 0.19 |
| **Height (cm)** | 1.64 ± 0.08 | 1.62 ± 0.08 | 0.21 |

Variables are presented as mean (SD) and standard deviation (DP).

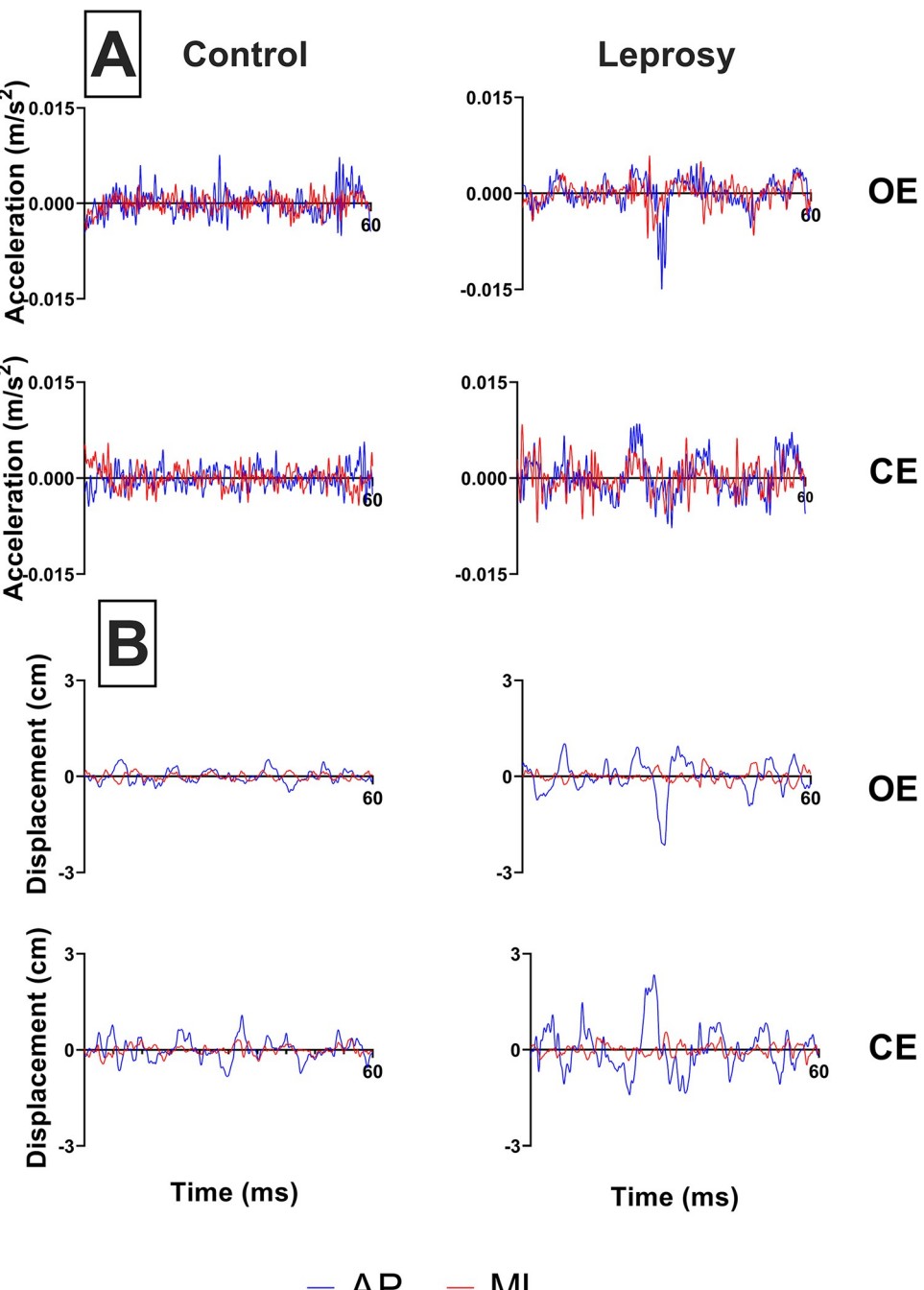

**Fig 1.** Inertial sensor stabilogram (A) and force platform stabilogram (B) of a representative subject from each group during the examination conducted in both conditions. Open eyes: OE; closed eyes: CE.

The scatter plots show the correlation between force platform measurements (displacement and area) and sensor acceleration data for two groups: Leprosy (yellow squares) and Control (blue squares). The plot indicates a positive correlation ($r = 0.58$, $p < 0.0001$) between total force platform displacement (x-axis) and sensor acceleration (y-axis). This suggests that higher displacement measured by the platform is associated with higher acceleration detected by the sensor for both groups. There is a positive correlation ($r = 0.61$, $p < 0.0001$) between the area

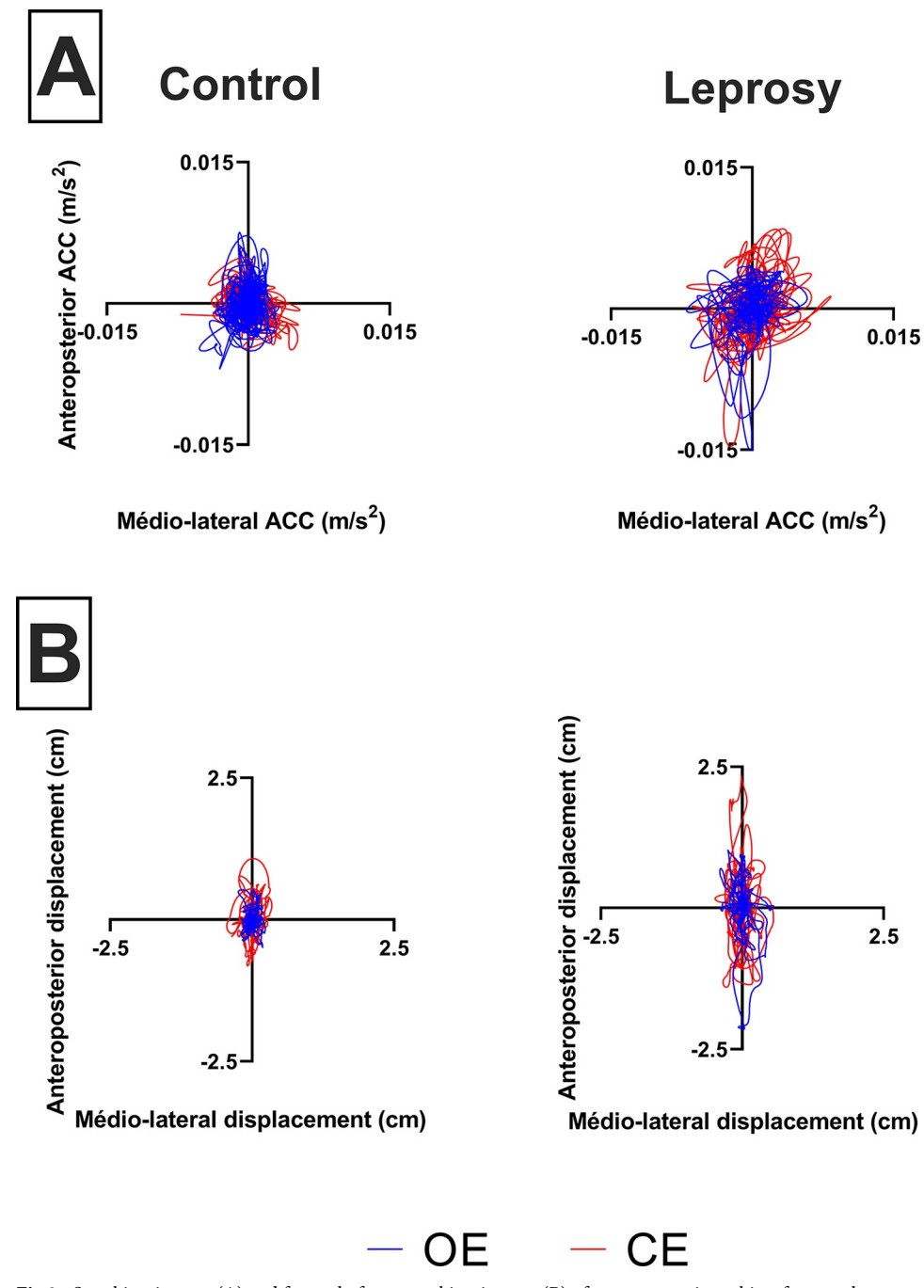

**Fig 2.** Statokinesiogram (A) and force platform statokinesiogram (B) of a representative subject from each group during the examination conducted under two conditions.

measured by the force platform and the sensor area. This means that as the area covered on the platform increases, the sensor area measurement also increases for both groups. The correlation between AP force platform displacement and AP sensor acceleration is positive ($r = 0.59$, $p < 0.0001$). This shows a strong relationship between the forward-backward movements detected by the platform and the sensor. The ML displacement plot shows a positive correlation ($r = 0.40$, $p < 0.0001$) between medio-lateral force platform displacement and ML

**Table 2. Quantitative parameters of the balance assessment per instrument.** The values represent median (interquartile range).

| | | Control group | Leprosy group | U/t,df | p-valor |
|---|---|---|---|---|---|
| **OE** | | | | | |
| TD | Sensor (m/s$^2$) | 4.69 (3.92;6.06) | 6.40 (4.65;8.04) | U = 280 | 0.01* |
| | Force platform (cm) | 589.9 (515.3;687.9) | 744 (585;1035) | U = 282 | 0.01* |
| AP$_{disp}$ | Sensor (m/s$^2$x10$^{-3}$) | 1.27 (1.2;1.69) | 1.87 (1.42;2.85) | U = 206 | 0.001* |
| | Force platform (cm) | 0.20 (0.18;0.24) | 0.24 (0.19;0.35) | U = 306 | 0.03* |
| ML$_{disp}$ | Sensor (m/s$^2$x10$^{-3}$) | 1.26 (0.96;1.65) | 1.50 (0.92;2.21) | U = 382 | 0.32 |
| | Force platform (cm) | 0.09 (0,07;0.10) | 0.12 (0.09;0.19) | U = 238 | 0.001* |
| AREA | Sensor (m/s$^2$x10$^{-4}$) | 0.28 (0.19;0.39) | 0.35 (0.26;0.72) | U = 301 | 0.03* |
| | Force platform (cm) | 0.36 (0.25;0.51) | 0.50 (0.33;1.07) | U = 267 | 0.01* |
| **CE** | | | | | |
| TD | Sensor (m/s$^2$) | 4.96 (3.81;6.14) | 6.36 (4.54;9.51) | U = 266 | 0.01* |
| | Force platform (cm) | 750.2 (554.6;919.9) | 991.3 (763.6;1319) | T = 3.712 Df = 58 | 0.001* |
| AP$_{disp}$ | Sensor (m/s$^2$x10$^{-2}$) | 0.13 (0.10;0.16) | 0.20 (0.15;0.30) | U = 217 | 0.001* |
| | Force platform (cm) | 0.27 (0.21;0.34) | 0.34 (0.28;0.43) | T = 2.905 Df = 58 | 0.05* |
| ML$_{disp}$ | Sensor (m/s$^2$x10$^{-2}$) | 0.12 (0.10;0.16) | 0.14 (0.09;0.21) | U = 395 | 0.423 |
| | Force platform (cm) | 0.10 (0.07;0.13) | 0.15 (0.10;0.25) | U = 222 | 0.001* |
| AREA | Sensor (m/s$^2$x10$^{-4}$) | 0.30 (0.19;0.51) | 0.51 (0.28;0.98) | U = 275 | 0.01* |
| | Force platform (cm) | 0.48 (0.28;0.83) | 0.92 (0.60;2.15) | U = 228 | 0.001* |

Open eyes: OE; closed eyes: CE. Total displacement: TD; Anteroposterior: AP$_{disp}$; Mediolateral: ML$_{disp}$. U: Mann Whitney; T: T Student; DF: Frequency Distribution.
*p < 0.05

sensor acceleration. Although still significant, this correlation is weaker compared to the total and AP displacements.

Table 3 provides a detailed description of the values depicted in Fig 3, demonstrating the correlation between acceleration and simple reach achieved through the Spearman correlation coefficient. (Table 3)

## 4. Discussion

The present study aimed to validate the application of inertial sensors for patients with leprosy by establishing a correlation with the outcomes obtained from a force platform. We hypothesized that leprosy patients would exhibit larger COP oscillations, as well higher COM accelerations and that the inertial sensor would be able to present the difference between groups in a correlated manner with the force platform, providing clinical and concurrent validity for its use.

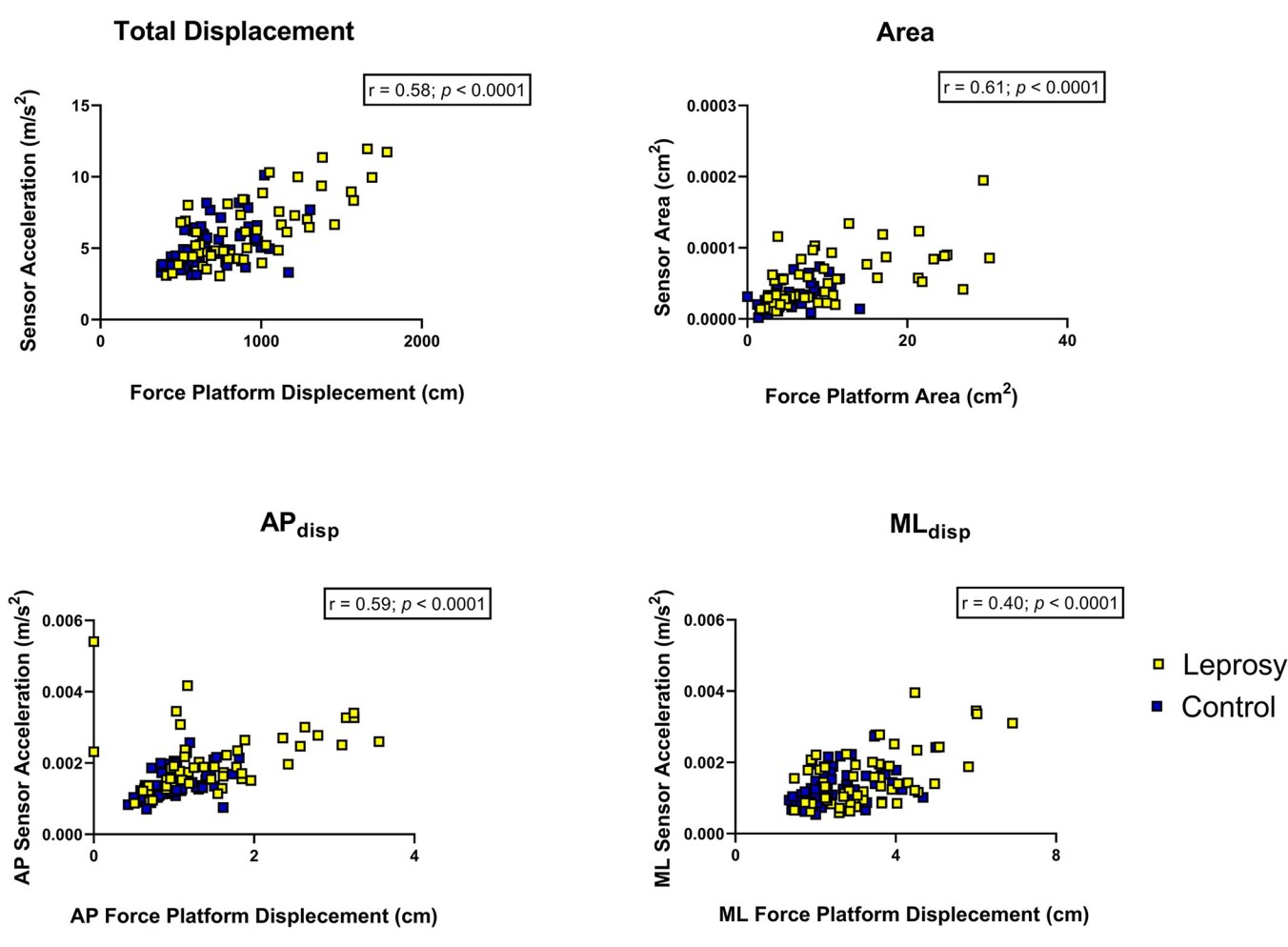

**Fig 3. Correlation between the force platform and the sensor.**

Our results partially confirmed the first hypothesis, regarding clinical validity, as not all differences between the groups, found in the COP measurements, were confirmed by the COM measurements. In the force platform assessments, participants with multibacillary leprosy

**Table 3. Correlation description.**

| | | OE | | | CE | | |
|---|---|---|---|---|---|---|---|
| | | r | IC | P value | r | IC | P value |
| TD | Control group | 0.509 | 0.171 | 0.004* | 0.416 | 0.055 | 0.022* |
| | Leprosy group | 0.435 | 0.036 | 0.029* | 0.788 | 0.580 | <0.000* |
| APdisp | Control group | 0.583 | 0.271 | 0.000* | 0.358 | -0.012 | 0.051 |
| | Leprosy group | 0.320 | -0.079 | 0.103 | 0.564 | 0.232 | 0.001* |
| MLdisp | Control group | 0.362 | -0.016 | 0.053 | 0.617 | 0.313 | 0.000* |
| | Leprosy group | 0.423 | 0.021 | 0.035* | 0.574 | 0.245 | 0.001* |
| AREA | Control group | 0.537 | 0.186 | 0.003* | 0.495 | 0.153 | 0.005* |
| | Leprosy group | 0.566 | 0.208 | 0.003* | 0.534 | 0.190 | 0.003* |

Open eyes: OE; closed eyes: CE. Total displacement: TD; Anteroposterior: $AP_{disp}$; Mediolateral: $ML_{disp}$. R: Spearman; IC: Confidence interval; T Student; *$p < 0.05$

exhibited greater amplitudes compared to the control group in all measurements. This was statistically confirmed by the sensor, except for $ML_{disp}$. Besides this, there was moderate to large correlation between the force platform and the sensor, what confirms the second hypothesis, regarding concurrent validation.

Our COP results align with the scarce existing literature that compared healthy individuals with leprosy patients, which also demonstrated greater oscillations in the group with the pathology [8,9]. Da Cruz Junior et al [8] had previously demonstrated impaired balance in the leprosy group through force platform evaluation and correlated these findings with increased plantar pressure in the forefoot region. The study by Viveiro et al [24] revealed that the balance control of these individuals, assessed through a force platform, manifests a greater oscillation area and higher COP velocity, particularly in the forward-backward direction of the ankle and Cordeiro et al [10] did not found statistical evidence that patients with abnormal plantar sensibility presented postural balance control disparities compared to control group.

In addition to the recognized issue of sensory loss, which can affect balance control [25], other mechanisms may also play a significant role in motor control alterations in leprosy patients. This impairment may also be associated with neural [26] infection and the inflammatory response triggered by Schwann cells, which cause degeneration of peripheral nervous system fibers [27].

The concept of "silent neuropathy" or "quiet nerve paralysis" is particularly relevant, as it refers to the impairment of sensory and motor functions without the classic clinical signs of leprosy, such as skin lesions, nerve tenderness, pain, or paraesthesia. This condition indicates that leprosy can lead to underlying nerve damage and dysfunction, even when overt clinical symptoms are absent.

Therefore, it is essential to consider other potential factors that could impact balance control in leprosy patients, including autonomic dysfunction, proprioceptive deficits, and the involvement of both myelinated and unmyelinated nerve fibers, as highlighted in the literature [28]. These factors can contribute to alterations in motor control and balance, emphasizing the need for comprehensive neurological assessment and early detection of subtle neuropathic changes to prevent the progression to more severe impairments and disabilities.

The balance of leprosy patients may be compromised regardless of gender; there are indications in the literature that men and women are affected in the same way by infectious diseases due to physiological issues, such as genetic interactions and sex hormones, which determine whether one sex will be more susceptible to infections, as well as differences in behavioral patterns, indicating greater exposure of either men or women to contagion factors [29].

Despite the COP parameters showing differences between the groups, the assessment of the COM acceleration with the sensor did not reveal $ML_{disp}$ as a variable confirming this difference. It is observed that postural control occurs through the weight distribution of the feet and their corresponding ground reaction forces, mainly through ankle adjustments in the antero-posterior direction [30]. This finding may explain our results, as it has been demonstrated that greater COP and COM in $AP_{disp}$, in Leprosy patients [9]. $ML_{disp}$, as previously mentioned, did not differ between the groups in the sensor measurements, and this was also the variable with the lowest correlation among the sensors, as observed in concurrent validation.

Moreover, although the present sensor had been previously validated in a study with a healthy population [22] and showed good correlations with the force platform, in this study, it did not demonstrate the same sensitivity in registering postural oscillations $ML_{disp}$.

The validation of the inertial sensor is extremely important, as it is a sensitive method that can guide its possible applicability in clinical settings, while also being cost-effective and easy to use. Despite the increasing evidence on the use of accelerometers for the assessment of static

balance in various population, this is the first study employing this technology in patients with multibacillary leprosy, what makes our results incomparable.

This study had the limitation of conducting only one acquisition with eyes open and one with eyes closed. Despite having a duration of 60 seconds, this aspect may be subject to questioning, especially when comparing between groups.

## 5. Conclusion

This research confirmed the postural control impairment in leprosy patients. Moreover, the inertial sensor can be applied in clinical settings due to its low cost and ease of use. However, it may require caution when measuring ML accelerations, since it did not assess the difference between groups demonstrated by the platform. The findings can contribute to public health by identifying low-cost tools to monitor the postural control of patients with multibacillary leprosy.

## Supporting information

**S1 Dataset. The S1 Dataset contains 41 entries with 16 columns, showing balance assessment measurements for control and healthy groups.** The parameters include Time Domain, Anterior-Posterior, Medial-Lateral, and Area measurements, with eyes open and closed for both groups. It provides comparative data on stability across different conditions. (XLSX)

## Author Contributions

**Conceptualization:** Amelia Pasqual Marques, Marília Brasil Xavier, Givago Silva Souza.

**Data curation:** Aymee Lobato Brito, Yuzo Igarashi, Luana Karine Resende Oliveira, Luciane Lobato Sobral.

**Formal analysis:** Yuzo Igarashi, Luana Karine Resende Oliveira, Luciane Lobato Sobral, Bianca Callegari.

**Funding acquisition:** Marília Brasil Xavier, Bianca Callegari.

**Investigation:** Aymee Lobato Brito, Yuzo Igarashi.

**Methodology:** Givago Silva Souza, Bianca Callegari.

**Project administration:** Bianca Callegari.

**Resources:** Marília Brasil Xavier.

**Software:** Yuzo Igarashi.

**Supervision:** Givago Silva Souza, Bianca Callegari.

**Validation:** Givago Silva Souza, Bianca Callegari.

**Visualization:** Marília Brasil Xavier, Givago Silva Souza, Bianca Callegari.

**Writing – original draft:** Aymee Lobato Brito, Amelia Pasqual Marques, Luana Karine Resende Oliveira, Givago Silva Souza, Bianca Callegari.

**Writing – review & editing:** Amelia Pasqual Marques, Givago Silva Souza, Bianca Callegari.

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
