## [Decision Letter · Decision Letter 0]

25 Apr 2024

Dear Dr. Callegari,

Thank you very much for submitting your manuscript "USE OF AN INERTIAL SENSOR AND A FORCE PLATFORM TO ASSESS STATIC BALANCE IN PARTICIPANTS AFFECTED BY MULTIBACILLARY LEPROSY" for consideration at PLOS Neglected Tropical Diseases. As with all papers reviewed by the journal, your manuscript was reviewed by members of the editorial board and by several independent reviewers. In light of the reviews (below this email), we would like to invite the resubmission of a significantly-revised version that takes into account the reviewers' comments. 

The manuscript needs revisions, especially in the presentation of the results and the conclusion. 

Please proceed with the revision and resubmit the manuscript for further analysis.

We cannot make any decision about publication until we have seen the revised manuscript and your response to the reviewers' comments. Your revised manuscript is also likely to be sent to reviewers for further evaluation.

Sincerely,

Susilene Maria Tonelli Nardi, Ph.D

Academic Editor

Ana LTO Nascimento

Section Editor

Dear authors.

The manuscript needs revisions, especially in the presentation of the results and the conclusion. 

Please proceed with the revision and resubmit the manuscript for further analysis.

Best wishes

Reviewer's Responses to Questions

**Key Review Criteria Required for Acceptance?**

**Methods**

-Are the objectives of the study clearly articulated with a clear testable hypothesis stated?

-Is the study design appropriate to address the stated objectives?

-Is the population clearly described and appropriate for the hypothesis being tested?

-Is the sample size sufficient to ensure adequate power to address the hypothesis being tested?

-Were correct statistical analysis used to support conclusions?

-Are there concerns about ethical or regulatory requirements being met?

Reviewer #1: The purpose of the research was clearly specified.

The population was well selected - although it must be admitted that it is small.

The discussion explains the answer to the research hypothesis.

Statistical calculations correct and nicely described.

comments:

The reference group is dominated by men and the study group is dominated by women - does this not affect the results in case of comparisons between groups?

Please add more information about the COM sensor to the article. What was recorded with it, what quantities, how filtered, how processed

Reviewer #2: The object of the study is to validate the use of the inertial sensor for leprosy patients through a correlation with the results from a force platform, not the evaluation of the leprosy and healthy subjects as stated by the author at the introduction section. 

 Also at the introcuction, the author shows references that imply that multibacilary leprosy patients have altered weight bearing disposition due to sensitivity loss, but the sample used in this study had not alteration in sensitivity, since they were all graded 0 (zero) disability, and those with altered sensitivity were graded 1. Therefore, the introduction does not justify the methods used. 

 It was not specifyed if the participants had any level of regular physical activity, since it would impact the results of the inertial sensors and balance control by the hip muscles, and it might explain why the results from MLdisp were not statisticaly different, as opposed to the other variables acquired.

**Results**

-Does the analysis presented match the analysis plan?

-Are the results clearly and completely presented?

-Are the figures (Tables, Images) of sufficient quality for clarity?

Reviewer #1: Results presented correctly. unreservedly.

comments:

What does adding graphs over time for one example person contribute to the article - please explain?

Reviewer #2: In Table 2, it was not specifyed what the value "U/t,df" stands for.

 The english grammar should be reviewed for the abstract and the results section. 

 The sensor validity part of the results section does not describe how the correlation between acceleration (m/s2) and simple range (cm) were made, and does not show data colected from "closed eys" situation. The adition of a table is strongly recommended to elucidade what is mainly visual on this part. 

The discussion section re-state that the plantar sensitivity alterations contribute to higher values of COP and COM for the leprosy subject. However, those subjects inluded in this study did not have any sensitivity alteration, which invalidade the choice of sample chosen, and opens up a new question as to why the results were different among the two groups. Is it possible that the degree of disability was not correctly colected?

**Conclusions**

-Are the conclusions supported by the data presented?

-Are the limitations of analysis clearly described?

-Do the authors discuss how these data can be helpful to advance our understanding of the topic under study?

-Is public health relevance addressed?

Reviewer #1: What COP and COM values are we talking about in the sentence:

"We hypothesized that COP and COM values would be higher in patients with Hansen's disease, and the sensor and platform would indicate group differences in a correlated manner."?

Reviewer #2: The limitations of analysis were described as the single acquisition of data with eyes oppend and closed, but I suggest that hte limitation should be that the leprosy sample does not have any sensitivity impairment. It muddles all the means and the discussion of the article

The conclusions of this study were presented within the Discussion section. It is strongly advised to sepparate them.

**Editorial and Data Presentation Modifications?**

Reviewer #1: Please include sentences explaining the issues raised in the comments sections in the body of the article.

Reviewer #2: (No Response)

**Summary and General Comments**

Reviewer #1: The subject of the article is the assessment of postural disorders in people AFFECTED BY MULTIBACILLARY LEPROSY. The research was carried out correctly with good reference to literature sources. The results, although simple, show differences in the measured parameters, which has a research aspect and the possibility of their use in clinical practice.

Reviewer #2: (No Response)

PLOS authors have the option to publish the peer review history of their article (what does this mean?). If published, this will include your full peer review and any attached files.

Reviewer #1: No

Reviewer #2: No
---

## [Decision Letter · Decision Letter 1]

30 Aug 2024

Dear Dr. Callegari,

Thank you very much for submitting your manuscript "USE OF AN INERTIAL SENSOR AND A FORCE PLATFORM TO ASSESS STATIC BALANCE IN PARTICIPANTS AFFECTED BY MULTIBACILLARY LEPROSY" for consideration at PLOS Neglected Tropical Diseases. As with all papers reviewed by the journal, your manuscript was reviewed by members of the editorial board and by several independent reviewers. In light of the reviews (below this email), we would like to invite the resubmission of a significantly-revised version that takes into account the reviewers' comments. 

Dear authors. 

Please make the necessary changes to the text of the manuscript based on the considerations of the reviewers. 

Authors can give reasons if they do not agree with the reviewer's suggestion. 

I would like to inform you that the title is appropriate.

Best wishes

We cannot make any decision about publication until we have seen the revised manuscript and your response to the reviewers' comments. Your revised manuscript is also likely to be sent to reviewers for further evaluation.

Sincerely,

Susilene Maria Tonelli Nardi, Ph.D

Academic Editor

Ana LTO Nascimento

Section Editor

Dear authors. 

Please make the necessary changes to the text of the manuscript based on the considerations of the reviewers. 

Authors can give reasons if they do not agree with the reviewer's suggestion. 

I would like to inform you that the title is appropriate.

Best wishes

Reviewer's Responses to Questions

**Key Review Criteria Required for Acceptance?**

**Methods**

-Are the objectives of the study clearly articulated with a clear testable hypothesis stated?

-Is the study design appropriate to address the stated objectives?

-Is the population clearly described and appropriate for the hypothesis being tested?

-Is the sample size sufficient to ensure adequate power to address the hypothesis being tested?

-Were correct statistical analysis used to support conclusions?

-Are there concerns about ethical or regulatory requirements being met?

Reviewer #2: The pursue for balance control in leprosy patients might be justified due to the lack of plantar sensitivity. However, the sample of this Work does not have A plantar sensitivity impairment ( being grade 0 for disability measures ). It's My suggestion to the authors to simply verify the balance control of the leprosy patient regardless of the sensitivity, And bring light to other possibilities of balance control for this population, on the discussion section.

Reviewer #3: Title is not appropriate. Please edit it 

the objectives of the study were clearly articulated with a clear testable hypothesis stated.

Design of this study is case-control not cross-sectional.

the population were clearly described and appropriate for the hypothesis being tested.

the sample size was sufficient to ensure adequate power to address the hypothesis being tested.

statistical analysis used was not appropriate. you should use two-way 2*2 repeated measure ANOVA to assess the main effects and interaction between group and postural condition for each balance parameters. Please calculate effect sizes for each statistical analysis performed.

 There are not concerns about ethical or regulatory requirements being met.

**Results**

-Does the analysis presented match the analysis plan?

-Are the results clearly and completely presented?

-Are the figures (Tables, Images) of sufficient quality for clarity?

Reviewer #2: Table 1 displays " muscle strength " as a demographical feature of the sample, but only for the leprosy participantes. Why was this subject important enough to be on the table, since it was not described at the methodology section, nor was mentioned at the introduction? And Why was the control Group not evaluated for muscle strength as well?

The correlation between platform and sensors should be better described in the text.

Reviewer #3: Please consider comment about statistical analysis described above and based on revise the Results section.

the figures (Tables, Images) are sufficient quality for clarity

**Conclusions**

-Are the conclusions supported by the data presented?

-Are the limitations of analysis clearly described?

-Do the authors discuss how these data can be helpful to advance our understanding of the topic under study?

-Is public health relevance addressed?

Reviewer #2: commentaries already given on " Methods" section

Reviewer #3: the conclusions are supported by the data presented.

the limitations of analysis are partly described.

the authors discusses how these data can be helpful to advance our understanding of the topic under study.

public health relevance is addressed

**Editorial and Data Presentation Modifications?**

Reviewer #2: No suggestions, other than the ones above

Reviewer #3: (No Response)

**Summary and General Comments**

Reviewer #2: The reference # 11 dos not cite the disability grade as described in this manuscript . Please review the need to cite this. reference

Reviewer #3: This study was aimed to evaluate the static balance of participants with multibacillary leprosy and healthy participants using a force platform and an inertial sensor, and to perform clinical and concurrent validation of the inertial sensor. This research confirmed the concurrent validity of the inertial sensor with the force platform and its clinical validation, demonstrating that this instrument can be applied in clinical settings due to its low cost and ease of use. Overall, the study is interesting, however there are some clarifications needed.

Comment#1

Title: Title is not appropriate. Please edit it 

Comment#2

Abstract. Please state Discussion more briefly and precisely. 

Comment#3

Methods. Design of this study is case-control not cross-sectional.

Comment#4

Please explain your justification for postural conditions selected to assess postural balance.

Comment#5

Please state justification for balance parameters assessed.

Comment#6

Please provide reference/references for this sentence “Our COP results align with the scarce existing literature that compared healthy individuals with leprosy patients, which also demonstrated greater oscillations in the group with the pathology.”

PLOS authors have the option to publish the peer review history of their article (what does this mean?). If published, this will include your full peer review and any attached files.

Reviewer #2: No

Reviewer #3: No
---

## [Editor Report · Decision Letter 2]

13 Sep 2024

Dear Dr. Callegari,

We are pleased to inform you that your manuscript 'USE OF AN INERTIAL SENSOR AND A FORCE PLATFORM TO ASSESS STATIC BALANCE IN PARTICIPANTS AFFECTED BY MULTIBACILLARY LEPROSY' has been provisionally accepted for publication in PLOS Neglected Tropical Diseases.

Best regards,

Susilene Maria Tonelli Nardi, Ph.D

Academic Editor

Ana LTO Nascimento

Section Editor

---

## [Editor Report · Acceptance letter]

19 Sep 2024

Dear Callegari,

We are delighted to inform you that your manuscript, "USE OF AN INERTIAL SENSOR AND A FORCE PLATFORM TO ASSESS STATIC BALANCE IN PARTICIPANTS AFFECTED BY MULTIBACILLARY LEPROSY," has been formally accepted for publication in PLOS Neglected Tropical Diseases.

Best regards,

Shaden Kamhawi

co-Editor-in-Chief

Paul Brindley

co-Editor-in-Chief
